# A shuttling-based two-qubit logic gate for linking distant silicon quantum processors

**Akito Noiri** [1] ✉, **Kenta Takeda** [1], **Takashi Nakajima** [1], **Takashi Kobayashi** [2], **Amir Sammak** [3,4], **Giordano Scappucci** [3,5] **& Seigo Tarucha** [1,2] ✉

Control of entanglement between qubits at distant quantum processors using a two-qubit gate is an essential function of a scalable, modular implementation of quantum computation. Among the many qubit platforms, spin qubits in silicon quantum dots are promising for large-scale integration along with their nanofabrication capability. However, linking distant silicon quantum processors is challenging as two-qubit gates in spin qubits typically utilize short-range exchange coupling, which is only effective between nearest-neighbor quantum dots. Here we demonstrate a two-qubit gate between spin qubits via coherent spin shuttling, a key technology for linking distant silicon quantum processors. Coherent shuttling of a spin qubit enables efficient switching of the exchange coupling with an on/off ratio exceeding 1000, while preserving the spin coherence by 99.6% for the single shuttling between neighboring dots. With this shuttling-mode exchange control, we demonstrate a two-qubit controlled-phase gate with a fidelity of 93%, assessed via randomized benchmarking. Combination of our technique and a phase coherent shuttling of a qubit across a large quantum dot array will provide feasible path toward a quantum link between distant silicon quantum processors, a key requirement for large-scale quantum computation.

Electron spins in silicon quantum dots attract a lot of interest as a platform of quantum computation with high-fidelity universal quantum control[1–3], long coherence time[4–6], capability of high-temperature operation[7,8], and potential scalability[9–12]. With recent technical advances, a densely-packed array of single-electron quantum dots works as a small-scale programmable quantum processor[1,2,13,14]. To scale up quantum computation by wiring to such dense qubit arrays and alleviating signal crosstalk, a quantum link is highly demanded that allows to manipulate entanglement between distant quantum processors in a sparse configuration[11,15]. A sizable exchange coupling required for two-qubit gates is, however, only achieved in qubits between nearest-neighbor quantum dots[1–3,6,12,13,16,17] as the coupling falls off exponentially with distance. Therefore two-qubit gates between distant quantum processors require coherent mediators such as microwave photons[18–21], empty and multi-electron quantum dots[22,23], and spin

chains[24]. Another approach uses electron shuttling[25–30] to physically move a qubit between quantum processors, bringing it wherever a two-qubit gate needs to be performed. However, a high-fidelity two-qubit gate in either approach is still challenging.

Here we propose and demonstrate a shuttling-based two-qubit gate which plays a key role in a quantum link between distant silicon quantum processors by electron shuttling. Figure 1a illustrates how this technique along with a coherent shuttling across a quantum dot array[26,27] can be used to interconnect two distant quantum processors via an empty quantum dot array, making a quantum link between them. More specifically, a qubit in one end of a quantum processor, which we call the moving qubit, is coherently moved to near the end of the other processor, where a sizable exchange coupling with a local qubit sitting there exists (Fig. 1a). Then the moving qubit is coherently shuttled back to the original quantum dot. In contrast to previous

[1]RIKEN Center for Emergent Matter Science (CEMS), Wako, Japan. [2]RIKEN Center for Quantum Computing (RQC), Wako, Japan. [3]QuTech, Delft University of Technology, Delft, The Netherlands. [4]Netherlands Organization for Applied Scientific Research (TNO), Delft, The Netherlands. [5]Kavli Institute of Nanoscience, Delft University of Technology, Delft, The Netherlands. ✉e-mail: akito.noiri@riken.jp; tarucha@riken.jp

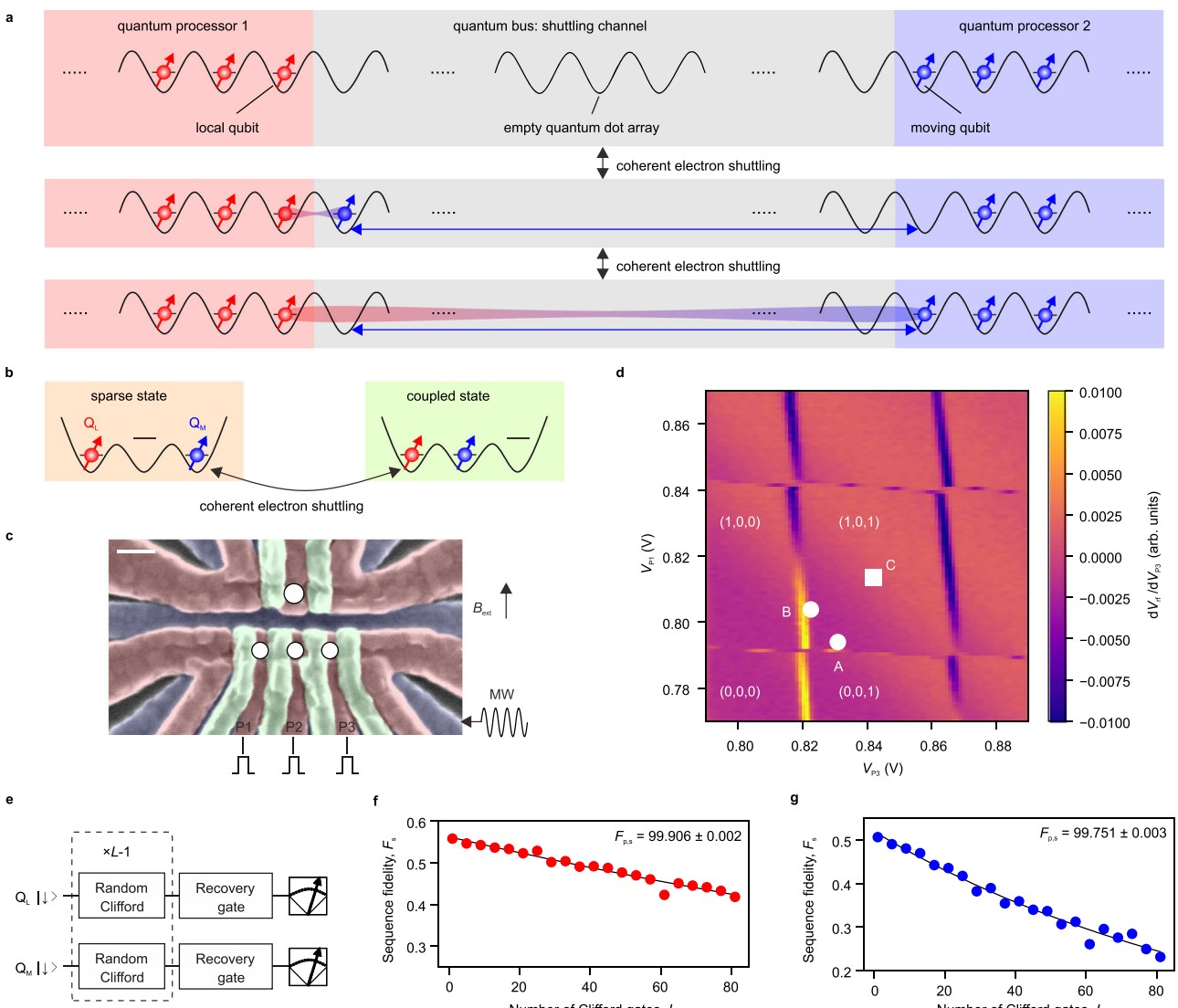

**Fig. 1 | Concept of experiment and qubit characterization. a** Concept of the experiment showing how a two-qubit gate between spin qubits at distant quantum processors is implemented to control entanglement between the qubits, making a quantum link between the quantum processors. The two-qubit gate is executed by coherent shuttling of the moving qubit to control the exchange coupling. An empty quantum dot array is used as a shuttling channel which works as a quantum bus. In the experiment, we use a tunnel coupled triple quantum dot containing two spin qubits in two quantum processors and a shuttling channel consisting of a single quantum dot as shown in **b**. **b** Two operation states used in this work. The local and moving qubits are located apart in the end quantum dots (sparse state) and they are coupled when the two qubits are located in the nearest-neighbor quantum dots (coupled state). **c** False color scanning electron microscope image of a device nominally identical to the one used in this work[1]. The white circles show the

position of the quantum dots hosting the qubits. The upper single electron transistor (shown by the large white circle) is used for radiofrequency-detected charge sensing[47,48]. The white scale bar shows 100 nm. An in-plane external magnetic field $B_{ext} = 0.45$ T is applied. **d** Charge stability diagram around the sparse state obtained by differentiating the charge sensing signal $V_{rf}$. White circles show the initialization and measurement conditions for $Q_L$ (labeled A) and $Q_M$ (labeled B). The white square (labeled C) shows the charge symmetry-point where single-qubit gates are implemented. **e** Quantum circuit for the single-qubit Clifford-based randomized benchmarking measurement used to produce (**f**) and (**g**). The same gate sequence is applied to both qubits simultaneously. **f**, **g** Single-qubit primitive gate fidelity characterized at the sparse state using Clifford-based randomized benchmarking for $Q_L$ (**f**) and $Q_M$ (**g**) (Methods). The uncertainty in the gate fidelities is obtained by a Monte Carlo method[1,38].

demonstrations of a controlled-phase (CZ) gate[2,3,6,13], our two-qubit gate between the local and moving qubits relies on dynamical switching of the exchange coupling by the shuttling processes. This technique will enable to implement the two-qubit gate between qubits at distant quantum processors when combined with shuttling across a long channel.

The experiment is performed in a tunnel coupled triple quantum dot hosting two qubits, a minimum setup to demonstrate the shuttling-based two-qubit gate. Initially, the local and moving qubits $Q_L$ and $Q_M$ are in the left and right dots, respectively, where parallel quantum processing with simultaneous single-qubit gates is performed. We refer to this configuration as the sparse state (Fig. 1b). The

negligible coupling between the qubits enables us to maintain the high fidelity of single-qubit gates while driving both qubits simultaneously. To perform a two-qubit gate, $Q_M$ in the right dot is shuttled to the center dot, and at the same time, the exchange coupling is turned on. We refer to this state as the coupled state (Fig. 1b). The shuttling-mode exchange switching allows us to efficiently control exchange coupling with an on/off ratio above 1000. By tuning an evolution time in the coupled state, we realize a CZ gate with a fidelity of 93%. Practically, a quantum link that can couple qubits separated by ~10 μm distance is useful for scaling up[15]. Along with the shuttling-based CZ gate, this requires high-fidelity coherent shuttling across a large quantum dot array. With a sufficiently large inter-dot tunnel coupling, we

demonstrate that 99.6% of the spin phase coherence is preserved in a single shuttling cycle. Then, challenges to be overcome include precise control of a large quantum dot array. A virtual gate technique is useful for tuning up such a quantum dot array in a scalable manner[25,31]. Furthermore, a recent demonstration of conveyer-mode shuttling[32] can decrease the number of control signals in a long-distance shuttling. In this approach, a qubit is shuttled by an electrostatically defined travelling potential created by an array of gate electrodes which are connected to one of the four control signal sources. Then, the number of control signals is independent of the length of shuttling channel, potentially reducing the complexity of controlling a long shuttling channel. With such technical advances, our technique can implement a quantum link between spin qubits at distant quantum processors that is useful for scaling up.

## Results

The device is fabricated on an isotopically enriched silicon/silicongermanium heterostructure. Three layers of aluminum gates create confinement potentials to define the quantum dots[9] (Fig. 1c). We operate this device in two charge configurations with two qubits: the coupled state (1,1,0) and the sparse state (1,0,1), where (l, m, n) denotes the number of electrons in the left (l), center (m) and right (n) dot. On top of the quantum dots, a cobalt micromagnet is fabricated to induce a magnetic field gradient required for electric-dipole spin resonance (EDSR) control of both qubits[33]. In addition, the field gradient makes a Zeeman energy difference of 403 MHz between the left and the center dot (Supplementary Fig. 1a, b). Compared to the Zeeman energy difference, an exchange coupling $J$ in a range of 1–10 MHz is small, and it shifts the energy levels defined by the Zeeman energy when the two spins are anti-parallel. This enables us to implement a CZ gate by simply turning on and off $J$[2,12,13].

We first demonstrate initialization, measurement, and single-qubit control of the spin qubits in the sparse state. White symbols in Fig. 1d show the gate voltage conditions used for the respective stages. Initialization and measurement are performed by energy-selective tunneling between quantum dots and their adjacent reservoirs[34,35]. Supplementary Fig. 1a, c demonstrates EDSR control of $Q_L$ and $Q_M$. The resonance frequencies differ by 733.4 MHz due to the micromagnet and this is large enough to control both qubits individually. The dephasing times $T_2^*$ are 3 and 4 μs for $Q_L$ and $Q_M$, which are enhanced by the echo sequence to 18 and 28 μs, respectively (Supplementary Fig. 2a–d). We also obtain the Rabi decay times long enough (>30 μs) for high-fidelity single-qubit gates (Supplementary Fig. 2e, f). We characterize the single-qubit gate fidelities by the simultaneous Clifford-based randomized benchmarking (Fig. 1e). We obtain high-fidelity single-qubit gates (single-qubit primitive gate fidelities of $F_{p,s} = 99.906 \pm 0.002\%$ for $Q_L$ and $99.751 \pm 0.003\%$ for $Q_M$ in Fig. 1f, g) even when the same gate sequence is applied to both qubits simultaneously, which shows that these qubits work as two independent single-qubit quantum processors.

Next, we demonstrate coherent shuttling of $Q_M$ using the right and center dots (Fig. 2a). The white symbols in Fig. 2b show the two gate voltage conditions for the coupled and the sparse states. The estimated inter-dot tunnel coupling $t_R$ between the dots is 20.2 GHz (Supplementary Fig. 4). After preparing $Q_M$ in the state of either spin-down or spin-up, we shuttle $Q_M$ back and forth by applying the pulse sequence shown in Fig. 2c and measure the final spin-up probability. Figure 2d shows that the initial spin polarization decays with the number of the shuttling cycles $n$. We extract the spin preservation fidelity per a shuttling cycle to be $F_d = 99.975 \pm 0.012\%$ for the spin-down state and $F_u = 99.971 \pm 0.007\%$ for the spin-up state, respectively. The preservation fidelity of the phase coherence is similarly evaluated by preparing $Q_M$ in the spin-down and -up superposition state and measuring the coherence decay (Fig. 2e). We obtain a coherence preservation fidelity per a shuttling cycle of $F_p = 99.62 \pm 0.05\%$ (Fig. 2f).

These fidelities are comparable to those reported in a silicon MOS quantum dot device[26]. This suggests that $Q_M$ can be shuttled over ~500 dots (distance of ~45 μm assuming a dot pitch of 0.09 μm) before the phase coherence decays by a factor of 1/e. We note that the phase of $Q_M$ shifts when it is shuttled across dots with different Zeeman energies that originate from the micromagnet-induced gradient field and a change in the interface roughness of the heterostructure across dots[36]. Since $t_R$ is sufficiently large for adiabatic shuttling of $Q_M$, this phase shift is a deterministic coherent phase shift which can be removed by a phase gate implemented by shifting phases of subsequent control microwave pulses in zero gate time[1,12,37].

Then, we demonstrate switching of the exchange coupling $J$ between $Q_L$ and $Q_M$ by shuttling $Q_M$. This allows us to implement a two-qubit gate between the $Q_L$ and $Q_M$ just by switching the operation states via coherent shuttling. To tune up $J$ in the coupled state, we tilt the energy levels of the left and center dots by the tilt voltage $V_{tilt}$ along the black axis in Fig. 3a. $J$ at the coupled state is evaluated by applying the quantum circuit shown in Fig. 3c with which $Q_M$ accumulates the controlled phase depending on the evolution time $t_{evol}$ at the coupled state (Fig. 3b). The π rotations for both qubits in the middle of the phase evolution decouple quasi-static noise[13]. Figure 3d shows $J$ and the decoupled dephasing times for both qubits as a function of $V_{tilt}$. While $J$ monotonically increases with increasing $V_{tilt}$, the decoupled dephasing times are barely affected between $V_{tilt} = 0$ V and $V_{tilt} = 0.012$ V and they start decreasing with increasing $V_{tilt}$ above 0.012 V. Therefore, we use $V_{tilt} = 0.012$ V with $J = 1.25$ MHz to implement the CZ gate at the maximum performance. On the other hand, we obtain a negligibly small $J$ of 0.9 kHz in the sparse state (Supplementary Fig. 6b). These results demonstrate that a more than one thousand switching ratio of $J$ is obtained by coherent electron shuttling.

We now use this shuttling-mode switching of $J$ to implement a CZ gate[13] between $Q_L$ and $Q_M$. The CZ gate is operated by tuning the evolution time in the coupled state to $t_{evol} = 1/2J = 0.4$ μs with single-qubit phase corrections made by shifting the phase of subsequent control pulses[12,13,37]. We use a decoupled CZ (DCZ) gate[12,13] to suppress dephasing during the controlled-phase evolution (Fig. 4a). To verify the construction of the DCZ gate, we measure the phase of $Q_M$ after the DCZ gate (Fig. 4b) using the quantum circuit shown in Fig. 4a. The obtained controlled phase is $1.00 \pm 0.01$ π from which we demonstrate an execution of the DCZ gate. We note that the CZ gate can be implemented by the DCZ gate followed by single-qubit gates acting on both qubits (Fig. 4c). From the results, we demonstrate that the CZ gate is appropriately operated between $Q_L$ and $Q_M$.

Finally, we execute two-qubit randomized benchmarking to characterize the CZ gate[38,39]. The blue circles in Fig. 4f show the averaged sequence fidelity $F_t$ (Methods) measured by the Clifford sequence shown in Fig. 4d. From the decay of the sequence fidelity, we extract a two-qubit Clifford gate fidelity $F_C = 88.02 \pm 0.06\%$ (Methods). The CZ gate fidelity is characterized with an additional measurement (Fig. 4e) where the CZ gate is interleaved between each randomly chosen Clifford gate. By comparing the decay of sequence fidelities between with (red circles in Fig. 4f) and without the interleaved CZ gates, we extract the CZ gate fidelity of $F_{CZ} = 92.72 \pm 0.18\%$ (Methods). The obtained fidelity is mostly limited by dephasing due to the slow controlled-phase accumulation of 0.4 μs compared to the decoupled dephasing times of ~7 μs (Supplementary Fig. 8). Application of a barrier gate pulse in addition to the shuttling pulse would further improve the CZ gate fidelity by increasing $J$ around the charge-symmetry point (Supplementary Note 1). In addition, the fluctuations of EDSR resonance frequencies during the data acquisition contribute to the obtained infidelity of the CZ gate. We calibrate these parameters in every ~2 h and the total data acquisition takes ~10 h. More frequent auto-calibration during the measurement[37] would further improve the gate fidelity.

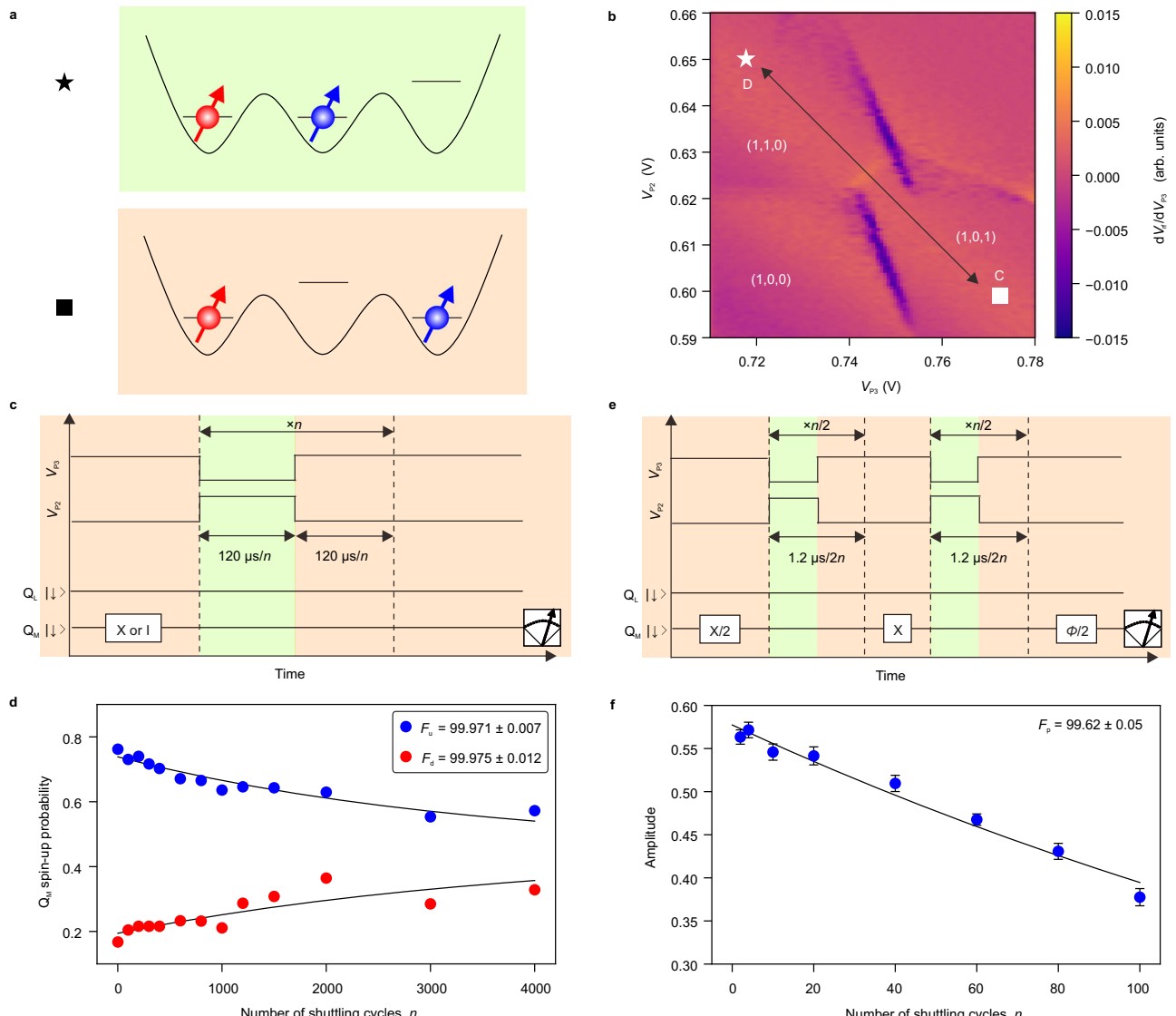

**Fig. 2 | Performance of coherent shuttling of $Q_M$. a** Energy diagram of the triple quantum dot at the gate voltage condition shown in the white symbols in (**b**). **b** Charge stability diagram around the (1,0,1) (sparse state) and (1,1,0) (coupled state) charge states. **c** Pulse sequence used to measure the spin-up probability depending on the number of shuttling cycles $n$ shown in (**d**). We use the charge symmetry-point at the coupled state. To eliminate the spin relaxation effect depending on time, the total time spent at each dot is fixed at 120 μs independent of $n$. **d** Spin-up probability after repeated shuttling cycles. We extract fidelities of the spin polarization preservation during a single back and forth shuttling cycle as $F_d$ (prepared in spin-down) and $F_u$ (prepared in spin-up) from fitting the data with an

exponential decaying function (black curves). The errors represent the estimated standard errors for the best-fit values. **e** Pulse sequence used to measure the amplitude in (**f**). A π rotation in the middle of the repeated shuttling cycles decouples quasi-static noise[26]. In addition, to eliminate the dephasing effect depending on time, the total time spent at each dot is fixed at 1.2 μs independent of $n$. **f** Number of shuttling cycles dependence of preservation of the spin phase coherence. The phase of the final π/2 rotation is varied to extract the oscillation amplitudes. We obtain the preservation fidelity $F_p$ of the spin phase coherence by fitting the data with an exponential decaying function. The errors represent the estimated standard errors for the best-fit values.

## Discussion

We also emphasize that the shuttling-mode exchange switching is beneficial for local qubit operations. A high-fidelity two-qubit gate requires large (~10 MHz) exchange coupling for a short gate time[1–3]. Except when operating the two-qubit gate, on the other hand, the coupling must be strictly turned off to below ~10 kHz to maintain the demonstrated high fidelity[5,40] of single-qubit gates (Supplementary Fig. 9). This is because the residual coupling induces a qubit energy shift conditional on neighboring qubit states, which decreases the single-qubit gate fidelity[2,12]. Typical residual coupling is a few tens of kHz[2,3] in the conventional scheme where the exchange coupling is switched by tilting the energy levels of quantum dots[6,41,42] and/or by modifying the potential barrier between quantum dots[16,17,43–45]. The shuttling-mode operation naturally enables to switch the exchange

coupling with a high enough on/off ratio of above 1000. We note that controllability of the coupling by the conventional schemes has been improved recently to the on/off ratio of 1000 in an advanced device structure[46] but an even larger on/off ratio may be required for further enhancing the gate fidelity. The shuttling-mode exchange switching can be used together with the conventional technique to improve the exchange controllability and thus favorable not only for linking distant quantum processors but also for implementing high-fidelity local qubit operations.

In summary, we demonstrate a CZ gate between silicon spin qubits by coherent shuttling of one of the qubits for linking distant quantum processors. The coherent shuttling allows us to shuttle a qubit while preserving its spin phase by 99.6% and simultaneously switch on and off the exchange coupling. The shuttling-mode

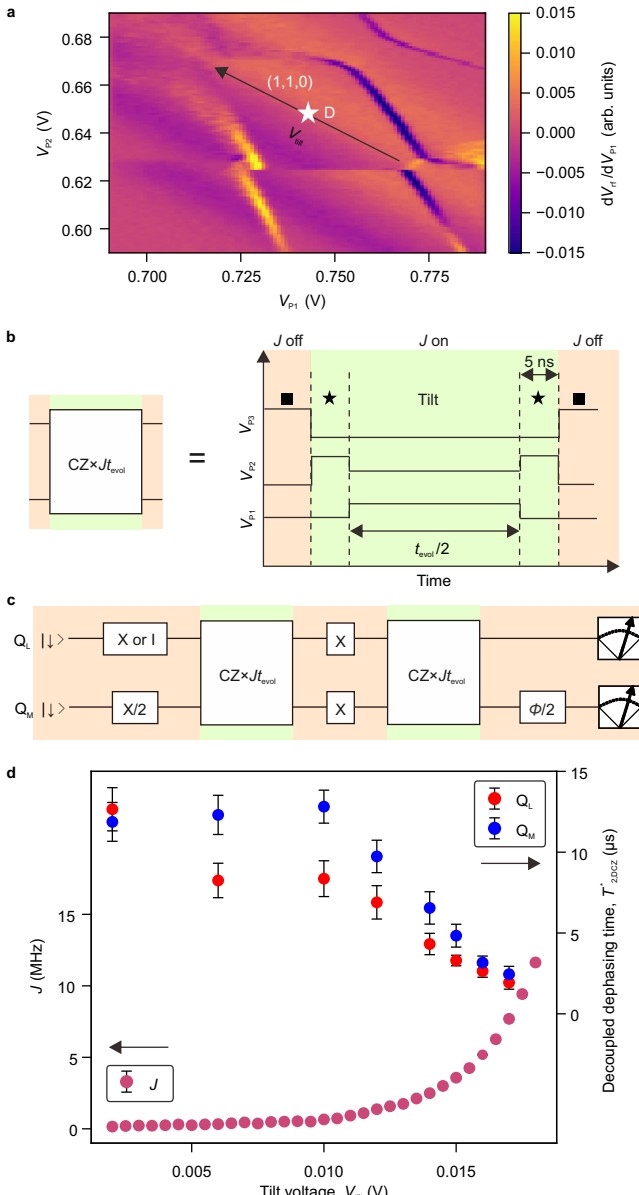

**Fig. 3 | Exchange coupling switching by coherent shuttling of $Q_M$. a** Charge stability diagram around the coupled state measured as a function of the P1 and P2 gate voltage. The black arrow and the white star show the tilt voltage axis and its origin ($V_{tilt} = 0$ V), respectively in (**d**). **b**, **c** A detailed voltage pulse sequence (**b**) and quantum circuit (**c**) used for measurement in (**d**). A wait time of 5 ns at $V_{tilt} = 0$ V (the charge symmetry-point) is inserted before and after a tilting voltage pulse to avoid unintentional charge transition during switching of the exchange coupling. **d** Operation point dependence of $J$ and the decoupled dephasing times for both qubits in the coupled state. A phase of the final $\pi/2$ rotation for $Q_M$ is varied to obtain the phase accumulation in $Q_M$. By comparing the phase of $Q_M$ in two different conditions of $Q_L$ prepared in spin-down and -up, we obtain the controlled-phase $2\pi J t_{evol}$ as a function of the total evolution time $t_{evol}$, from which we extract $J$. We find that $J$ is ~0.1 MHz at $V_{tilt} = 0$ V which is limited by a small tunnel coupling $t_L$ between the left and center dots (Supplementary Note 1). This is too small to implement a CZ gate with decoupled dephasing times of ~10 μs. $J$ can be enhanced by increasing $V_{tilt}$. The decoupled dephasing time of $Q_M$ is obtained from the exponential decay of the oscillation amplitude of spin-up probability as a function of the phase of the final $\pi/2$ rotation for $Q_M$ (Supplementary Fig. 8). Single-qubit gates for $Q_L$ and $Q_M$ are swapped to measure the decoupled dephasing time of $Q_L$. The decoupled dephasing times are longer than those obtained without decoupled sequence as shown in Supplementary Fig. 5b. The errors represent the estimated standard errors for the best-fit values.

exchange switching allows us to implement the CZ gate with a fidelity of 93% accompanied with a high on/off ratio of more than one thousand. Even higher gate fidelity will be achieved by an additional barrier gate pulse. These results demonstrate key technologies for a shuttling-based quantum link between distant quantum processors and thereby open a path to realization of large-scale spin-based quantum computation.

## Methods

### Measurement setup

The sample is cooled down in a dry dilution refrigerator (Oxford Instruments Triton) to the electron temperature of ~60 mK. The dc gate voltages are supplied by a 24-channel digital-to-analog converter (QDevil ApS QDAC), which is low-pass filtered at a cutoff frequency of 800 Hz. The voltage pulses applied to the P1, P2, and P3 gate electrodes are generated by an arbitrary waveform generator (Tektronix AWG5014C). The output of the arbitrary waveform generator is low-pass filtered at a cutoff frequency of 100 MHz, which limits the time required for the electron shuttling to ~3 ns. By inserting a ramp time for the shuttling pulse, we find that the preservation fidelity of spin phase coherence monotonically decreases with increasing the ramp time. Therefore, we omit the ramp time all through the experiments. The EDSR microwave pulses are generated using an I/Q modulated signal generator (Anritsu MG3692C with a Marki microwave MLIQ-0218 I/Q mixer) and applied to the bottom screening gate. The I/Q modulation signals are generated by another arbitrary waveform generator (Tektronix AWG70002A) triggered by the arbitrary waveform generator used for generating the gate voltage pulses.

### Sequence fidelity and gate fidelity extraction in randomized benchmarking

The sequence fidelity of single-qubit randomized benchmarking is obtained by the following procedure[1,4,12,13]. We measure two data sets of spin-up probability $P_\uparrow(L)$ and $P'_\uparrow(L)$ as a function of the number of Clifford gates $L$. Here, the recovery Clifford gate is chosen so that the final ideal state is spin-up for $P_\uparrow(L)$ and spin-down for $P'_\uparrow(L)$. Then the sequence fidelity $F_s(L)$ is obtained from $F_s(L) = P_\uparrow(L) - P'_\uparrow(L) = A_s p_s^L$, where $p_s$ is the depolarizing parameter and $A_s$ is the constant which absorbs the state preparation and measurement errors. We average 24 random sequences, each of which are repeated 1000 times to measure $F_s(L)$. The Clifford gate fidelity $F_{C,s}$ is obtained by $F_{C,s} = (1 + p_s)/2$. Since a Clifford gate contains 1.875 primitive gates on average, we extract the primitive gate fidelity $F_{p,s}$ as $F_{p,s} = 1 - (1 - F_{C,s})/1.875$.

Similarly, the sequence fidelity of two-qubit randomized benchmarking is extracted by the following procedure[1]. We measure $P_{\uparrow\uparrow}(L)$ ($P'_{\uparrow\uparrow}(L)$) as a function of $L$ with the recovery Clifford gate chosen so that the final ideal state is spin-up (spin-down) for both qubits. Here $P_{\uparrow\uparrow}$ and $P'_{\uparrow\uparrow}$ is the joint probability of spin-up in both qubits. Then the sequence fidelity $F_t(L)$ is extracted from $F_t(L) = P_{\uparrow\uparrow}(L) - P'_{\uparrow\uparrow}(L) = A_t p_t^L$, where $p_t$ is the depolarizing parameter and $A_t$ is the constant which absorbs the state preparation and measurement errors. We average 50 random sequences each of which are repeated 2000 times to measure $F_t(L)$. The two-qubit Clifford gate fidelity is obtained by $F_C = (1 + 3p_t)/4$.

The CZ gate fidelity is obtained as follows[1,38]. We first measure $F_t(L)$ by applying random Clifford gates (Fig. 4d) and obtain the depolarizing parameter $p_{ref}$ as a reference. We also measure $F_t(L)$ by applying the CZ gate between each random Clifford gates (Fig. 4e) and obtain the depolarizing parameter $p_{CZ}$. Then we extract the CZ gate fidelity as $F_{CZ} = (1 + 3p_{CZ}/p_{ref})/4$.

The errors of the gate fidelities are obtained by a Monte Carlo method[1,38]. We fit the resulting fidelity distribution by the Gaussian distribution and extract its standard deviation.

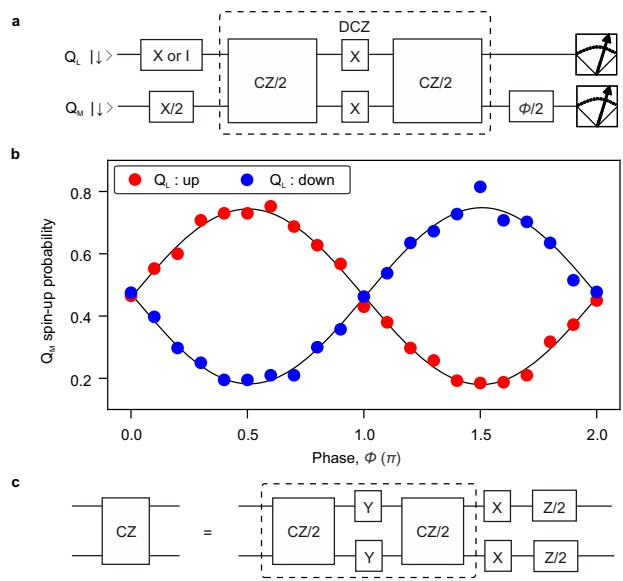

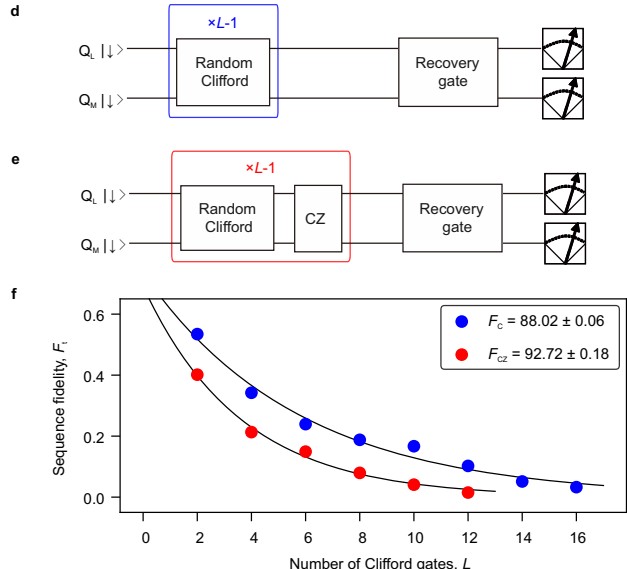

**Fig. 4 | Calibration and characterization of CZ gate. a** Quantum circuit used for calibrating the controlled-phase accumulation in $Q_M$ during the DCZ gate. **b** Accumulated phase on $Q_M$ in the DCZ gate operation when $Q_L$ is prepared in spin-down (blue) and -up (red) measured using the circuit shown in (**a**). Here, an unconditional phase accumulation of $0.04\pi$ for $Q_M$ is compensated by shifting the phase of the final $\pi/2$ rotation[12,13,37]. At the same time, $Q_L$ also acquires an unconditional phase of $0.065\pi$ (Supplementary Fig. 7b). We use $V_{tilt} = 0.012$ V at the coupled state where $J = 1.25$ MHz. **c** Quantum circuit for constructing the CZ gate from the DCZ gate and single-qubit gates. Here, we use Y gates acting on both qubits to implement the DCZ gate (inside the dashed square) instead of X gates used in (**a**). The single-qubit phase gates are implemented by changing phases of the subsequent control pulses[1,12,37]. **d, e** Quantum circuit for the two-qubit Clifford-based randomized benchmarking measurement without (**d**) and with (**e**) inter-leaved CZ gates, respectively. The two-qubit Clifford group has 11,520 elements all of which can be constructed from the combinations of CZ gates and single-qubit gates acting on both qubits[38,39]. **f** The two-qubit Clifford gate fidelity and the CZ gate fidelity extracted by the randomized benchmarking measurement (Methods). The uncertainty in the gate fidelities is obtained by a Monte Carlo method[1,38].

## Data availability

The data that support findings in this study are available from the Zenodo repository at https://doi.org/10.5281/zenodo.7033594.

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

## Acknowledgements

We thank the Microwave Research Group in Caltech for technical support. This work was supported financially by Core Research for Evolutional Science and Technology (CREST), Japan Science and Technology Agency (JST) (JPMJCR15N2 and JPMJCR1675), MEXT Quantum Leap Flagship Program (MEXT Q-LEAP) grant Nos. JPMXS0118069228, JST Moonshot R&D Grant Number JPMJMS226B, and JSPS KAKENHI grant Nos. 16H02204, 17K14078, 18H01819, 19K14640, and 20H00237. T.N. acknowledges support from JST PRESTO Grant Number JPMJPR2017.

## Author contributions

A.N. and T.N. conceived the project. A.N. and K.T. fabricated the device and performed the measurements. T.N. and T.K. contributed the data acquisition and discussed the results. A.S and G.S developed and supplied the $^{28}$silicon/silicon-germanium heterostructure. A.N. wrote the manuscript with inputs from all co-authors. S.T. supervised the project.

## Competing interests

The authors declare no competing interests.
