## [Peer Review File · Nature Communications]

REVIEWER COMMENTS

Reviewer #1 (Remarks to the Author):

Coherent shuttling of the spin qubits offers connectivity beyond the nearest neighbor quantum dot and is, therefore, considered to be a great advantage of gate-defined semiconducting qubits in reaching quantum advantage. This manuscript by A. Noiri et al. reports on realizing a two-qubit (controlled-phase) gate between distant electron spin qubits in Si/SiGe via coherent spin shuttling. The experiment is performed in a nanofabricated triple-quantum-dot-device in the presence of a micromagnet (nominally identical to the setup used in Ref. [1]) as a minimum setup in realizing a quantum link between distant spin qubits.

The authors first demonstrate the initialization, measurement and single-qubit gates in the 'sparse state' where the two spin qubits are separated by an empty quantum dot. This is followed by demonstrating coherent shuttling of a 'moving qubit' to the middle empty dot. The fidelities for preserving spin polarization and phase coherence during one shuttling cycle are found and turned out to be comparable to those reported before in Ref. 25 in a silicon MOS quantum device (however, the work in Ref. 25 is done in the absence of a B-field gradient generated by a micromagnet.) Importantly, the authors then demonstrate the switching of the exchange coupling between the local and moving qubit by shuttling. This, in turn, allows the implementation of a two-qubit gate by the coherent shuttling. In order to find the optimal parameters for gate implementation, in Fig. 3 the exchange coupling and decoupled dephasing times are found as a function of the tilt voltage where an optimal V_{tilt} is identified. The average gate fidelity at this optimal voltage is found to be ~ 0.93 from randomized benchmarking. One other interesting achievement is switching the exchange coupling with on/off ratio above 1000; this can establish coherent shuttling as a way to achieve high fidelity local qubit operation via suppressing the cross-talk.

The paper studies a subject that is very timely and is of great importance to the community working on silicon-based quantum processors. The experimental results presented are solid and can bring value and insight to the community. The manuscript is well written and has a clear structure. As such, I believe the work deserves publication in Nature Communication. However, I very much hope that the authors can take the following into consideration:

The reported average gate fidelity of 0.93 indicates much may need to be done in order to improve the fidelity. In that sense, error budgeting and a clear understanding of limiting factors are essential. The authors write "The obtained fidelity is mostly limited by dephasing due to the slow controlled-phase accumulation of $0.4 \mu\text{s}$ ". However, it would be really appropriate if the authors could elaborate more, at least qualitatively, on the different errors that would give rise to infidelity in the

current study. Also, can the authors provide some estimation about the improvement of gate fidelity by an additional barrier gate pulse?

I note that even if coherent errors account for a small fraction of the total error rate, there can be a significant difference between the worst-case infidelities and the average-case infidelity [see, e.g., J. Wallman et al., PRA 94, 052325 (2016) and A. Hashim et al., PRX 11, 041039 (2021)]. The authors write, “We note that the phase of Q_M shifts when it is shuttled across dots with different Zeeman energies, but this phase shift can be removed by a phase gate implemented by software”. I guess this is an attempt to avoid/suppress coherent errors; is that correct? I think in the present study, the difference between Zeeman energies across the dots is (largely) due to the B-field gradient. However, I think it is worth mentioning that change of the Si/SiGe interface roughness across different dots also contributes to this due to the change of the valley splitting as well as change in the effective strength of the spin-orbit interaction [see, e.g., A. Hosseinkhani et al., PRB 104, 085309 (2021)].

Reviewer #2 (Remarks to the Author):

The paper by Noiri et al. describes the implementation of a shuttling-based exchange gate protocol, based on a triple quantum dot sample in silicon and the use of a technique to swap spin states between neighboring quantum dots. Quite a few of the techniques used in this paper have recently been introduced by the same authors in Ref. 1.

The manuscript is detailed and well written, the analysis of the measurements is sound, and the described experiments constitute a step into a promising direction. Especially, the “more than one thousand switching ratio of J ” obtained is very impressive. I also agree with the authors that this method of switching the exchange gate on and off may be generally useful when they say “shuttling-mode exchange switching is beneficial for local qubit operations” and they should probably highlight this point earlier in the paper.

I am however not completely convinced that the experiment at its current stage constitutes an actual demonstration of “the CZ gate between silicon spin qubits at distant quantum processors” as the authors claim in the conclusion.

A weakness of the manuscript is that the authors select an absolute minimal configuration to demonstrate the technique, namely a linear triple quantum dot where they shuttle back and forth.

Using a series of empty dots as has been demonstrated previously (Mills. et. al., Ref. 26) would have added to the impact and solidified claims about the preservation of the gate fidelity over larger distances and a more varied and realistic disorder landscape. The first panel of Figure 1, a schematic, especially seems to make explicitly the claim of a long length channel, while the experiment itself is in a linear triple dot, on which similar experiments of similar or greater complexity have been performed previously. For example, the navigation in the gate spaces of longer quantum dot chains could in itself constitute a significant challenge to the protocol and I would therefore suggest shuttling over an empty chain of three to four dots would perhaps have been an actual minimal example or demonstration.

The dependence of the spin preservation fidelity on length and magnetic field gradient inhomogeneities from the cobalt micromagnet would also have been useful to see, and assess. The authors acknowledge that the “the phase of QM shifts when it is shuttled across dots with different Zeeman energies” but they dismiss it as “this phase shift can be removed by a phase gate implemented by software”. It is not clear how much of a roadblock, or an overhead, this is to the protocol and it would be nice to get a comment from the authors on this.

1) Could the authors comment on the effect of gate crosstalk on the fidelity of this protocol, especially if other qubits are around, or if the shuttling needs to be performed on longer length scales? I would have expected this to be something that needs to be compensated for, especially with the overlapping-gate geometry.

2) The authors also comment, if I understand right, that the high fidelity of single qubit gates can be maintained without making corrections for cross talk when the qubits are in the “sparse” position? They say “with this condition, we also obtain a high-fidelity ($F = 99.906 \pm 0.002\%$ for QL and $99.751 \pm 0.003\%$ for QM) single-qubit gates (Fig. 1f, g) with a Rabi frequency of 2.5 MHz even when the same gate sequence is applied to both qubits simultaneously”.

3) The manuscript also mentions that “Although the device is identical to the one used in ref. 1, we obtain a shorter dephasing time $T2^*$ for both qubits than those measured in ref. 1 possibly due to increased charge noise in the gate voltage condition used in this work”. It is not clear if the decreased $T2^*$ relates to the measurement in the sparse state or is simply due to changes in the experimental conditions between the two experiments; can the authors comment on this?

4) The paragraph (line 126) describing CZ gate implementation via a DCZ gate (and especially the equivalence via a Y and Z/2 gate) is not clear, and could be explained in a little more detail to convince the reader that the CZ is appropriately performed.

5) Was there a problem (device or experimental) due to which a symmetric or barrier-gate pulse was not used to operate the CZ gate (line 146) as the authors mention?

To conclude, the work is at a very high level and is very useful to the community; however due to the inequivalence of shuttling within a triple dot vs shuttling along longer distances, it is not clear to me that the claim of a long-range gate should be made. I feel that claim should be reserved for an actual demonstration of a long-range gate. Perhaps the paper can still be suitable for publication in this journal, given a change in the sections of the manuscript where authors have made claims of this type, such as “Here we overcome this problem by coupling spin qubits at distant quantum processors”, for example on line 23 and in the conclusion.

Lastly, a few typos:

Concluding paragraph: “we demonstrate a CZ gate” followed by “the coherent shuttling allows us to shuttle a qubit while preserving its spin phase by 99.6%”

Fig. 1 caption: “Concept of experiment and qubit characterization”

Reviewer #3 (Remarks to the Author):

Summary of the work

In this work, A. Noiri and colleagues demonstrate a coherent shuttling-based controlled phase (CZ) gate between two one-site-apart single-spin qubits (Loss-DiVincenzo qubit) in a triple quantum dot device in $^{28}\text{Si}/\text{SiGe}$. Specifically, they show coherent shuttling of an electron between the center (dot2) and right sites (dot3) by showing both transfers of basis and superposition states. They also implement coherent shuttling + echoed CZ gate with a fidelity of 93% deduced from two-qubit Randomized benchmarking protocol.

Evaluation

1. Strengths: I acknowledge the work's high-quality results. The authors' experiment is from one of the best quality silicon quantum information devices available to date. All single and two-qubit gate fidelity, coherent shuttling performance, and full two-qubit randomized benchmarking represent the state-of-the-art of the relevant research field. The manuscript is concise and generally well written except for a few points I describe below.

2. Clarification on novelty: Although the authors correctly cite previous demonstrations, I want to clarify that the demonstration of shuttling + CZ gate in combination is, and only this is, the novel aspect of the paper. Individual demonstration of coherent shuttling (both basis and superposition states) in ^{28}Si , high fidelity single-spin qubit gates, and echoed CZ gate and two-qubit Randomized benchmarking are all demonstrated either by other groups or by the authors previously. As such, the work's evaluation should be based on whether the current demonstration of shuttling + CZ gate indeed has a significant and promising impact on the long-range linking of qubits in distant quantum processors.

3. Weakness: I have main concerns as follows.

A. Major point: A credible (through RB) evaluation of 93% fidelity of shuttling-based CZ gate itself is not bad at all considering that it is the first demonstration of this kind. However, the work's main aim is to use coherent shuttling for linking distant qubits. Thus, in my opinion, a more correct evaluation should be as follows.

i. First perform conventional CZ gate between dot1 and dot2. That is conventional CZ when the electrons are sitting at dot1 and dot2 without shuttling. And evaluate through two-qubit RB.

ii. Then perform shuttling + CZ gate between dot1 and dot3 and do two-qubit RB. This is already demonstrated in the current work.

iii. Repeat (ii) with more than one shuttling before CZ. Compare the RB results with the result in (i), and concretely evaluate how much the shuttling process affects the reduction of CZ fidelity.

In short, my main concern is the absence of experiments A-(i) and A-(iii). For example, if I use the conventional CZ (coupled state CZ) fidelity of 99.5% previously demonstrated by the authors (Nature 601 338) in the same device but between dot2 and dot3, I should conclude that one shuttling process reduces CZ gate fidelity by $93/99.5 = 93.4\%$, which means the CZ fidelity drop down to 70%

with only five shuttling process. This is contrary to the authors' main claim; coherent shuttling is not promising for connecting distant qubits!

B. Major point: Related to A, the authors stress that the shuttling is beneficial for increasing on/off ratio of J. I agree with the authors. However, without the experiment described in A-(i) and (iii), the current paper's utility is rather for showing enhanced controllability over J through shuttling than using shuttling for connecting distant qubits. Note also that some of the barrier gates in the current device have limited J controllability. If the authors could manage to build a device with increased interdot distance, I would expect the same J controllability can be achieved without shuttling, making advantage of increasing J controllability with shuttling technique marginal.

C. Minor point: Coherent shuttling for linking distant qubits is a neat concept, but it does not reduce the complexity of the device architecture: for linking, one still needs to fabricate and use many control lines synchronously. Can authors put their opinion on this and discuss briefly?

D. Major-Minor mixed: I see in the Extended data 2 that T_2^* in the current device (28Si) is only marginally improved over the natural Si and authors ascribe to increased charge noise. Can authors put a bit more discussion on this? Why charge noise affects nominally charge-noise immune single spin qubits? Is it related to the interplay between increased charge noise and micromagnet? Related to my Major points, is this the reason that shuttle+CZ gate in the current work has lower fidelity 93% than Ref. 1 ? Not because of the shuttle process itself?

Overall, I acknowledge the work's highly qualified experimental results and the importance of linking distant qubits in the future. However, the current version of the manuscript lacks an important series of experiments as I described in 3.A. I understand that performing one two-qubit RB protocol takes a lot of time and there can be device-specific problems (barrier controllability, charge stability.. etc.) that prevents the execution of some of the additional experiments I suggest. However, these experiments are crucial for this paper, especially for talking about quantum links even though it is performed in a minimal set of triple quantum dots. Therefore, I would like to postpone my recommendation until I get a reply from the authors with some additional evidence or rebuttal to my points.

We thank the reviewers for their careful considerations on the manuscript. Our point-by-point responses to all comments from the reviewers and relevant changes in the manuscript are given below. All changes in the manuscript (except those of the reference numbers, supplementary figure numbers, and formatting changes) are highlighted in red. We believe that all the comments in the referees' reports are addressed and that our manuscript is now appropriate for publication in *Nature Communications*.

Response to Reviewer #1:

Comment #1

Coherent shuttling of the spin qubits offers connectivity beyond the nearest neighbor quantum dot and is, therefore, considered to be a great advantage of gate-defined semiconducting qubits in reaching quantum advantage. This manuscript by A. Noiri et al. reports on realizing a two-qubit (controlled-phase) gate between distant electron spin qubits in Si/SiGe via coherent spin shuttling. The experiment is performed in a nanofabricated triple-quantum-dot-device in the presence of a micromagnet (nominally identical to the setup used in Ref. [1]) as a minimum setup in realizing a quantum link between distant spin qubits.

The authors first demonstrate the initialization, measurement and single-qubit gates in the 'sparse state' where the two spin qubits are separated by an empty quantum dot. This is followed by demonstrating coherent shuttling of a 'moving qubit' to the middle empty dot. The fidelities for preserving spin polarization and phase coherence during one shuttling cycle are found and turned out to be comparable to those reported before in Ref. 25 in a silicon MOS quantum device (however, the work in Ref. 25 is done in the absence of a B-field gradient generated by a micromagnet.) Importantly, the authors then demonstrate the switching of the exchange coupling between the local and moving qubit by shuttling. This, in turn, allows the implementation of a two-qubit gate by the coherent shuttling. In order to find the optimal parameters for gate implementation, in Fig. 3 the exchange coupling and decoupled dephasing times are found as a function of the tilt voltage where an optimal V_{tilt} is identified. The average gate fidelity at this optimal voltage is found to be $\sim 0.93\%$ from randomized benchmarking. One other interesting achievement is switching the exchange coupling with on/off ratio above 1000; this can establish coherent shuttling as a way to achieve high fidelity local qubit operation via suppressing the cross-talk.

The paper studies a subject that is very timely and is of great importance to the community working on silicon-based quantum processors. The experimental results presented are solid and can bring

value and insight to the community. The manuscript is well written and has a clear structure. As such, I believe the work deserves publication in Nature Communication. However, I very much hope that the authors can take the following into consideration:

Response: We thank the reviewer for correctly understanding the importance of our work and thereby recommending publication in *Nature Communications*.

Comment #2

The reported average gate fidelity of 0.93 indicates much may need to be done in order to improve the fidelity. In that sense, error budgeting and a clear understanding of limiting factors are essential. The authors write “The obtained fidelity is mostly limited by dephasing due to the slow controlled-phase accumulation of 0.4 μs ”. However, it would be really appropriate if the authors could elaborate more, at least qualitatively, on the different errors that would give rise to infidelity in the current study. Also, can the authors provide some estimation about the improvement of gate fidelity by an additional barrier gate pulse?

Response: We thank the reviewer for the comment. First of all, we estimated the effect of dephasing during the controlled-phase accumulation on the gate fidelity. We measured the decay of the decoupled phase accumulation and found the dephasing times of ~ 7 and $10 \mu\text{s}$ for Q_L and Q_M , respectively with the exponential fitting [e.g., ref. 13]. Then, the effect of dephasing during the controlled-phase accumulation is roughly estimated as $e^{-(0.4/7)} = 94.5\%$. This suggests the most part of the error in our CZ gate comes from dephasing during the controlled-phase accumulation. Possible other important error sources are fluctuations of EDSR resonance frequencies which result in errors in single-qubit gates. Since our CZ gate has 4 single-qubit x- and y-rotations as shown in Fig. 4c (phase gates are omitted as they can be implemented by software without errors, see also response to Comment #3), errors in these gates contribute to infidelity of the CZ gate. Fidelities of single-qubit gates can be high as shown in Fig. 1f, g if they are measured right after calibration of EDSR resonance frequencies. In the measurement of the CZ gate fidelity in Fig. 4f, the total data acquisition time is ~ 10 hours and we calibrate these parameters every ~ 2 hours, which may not be frequent enough to maintain the high fidelities in the single-qubit gates obtained in Fig. 1f, g. More frequent auto-calibration of these parameters during the measurement would further improve the CZ gate fidelity.

In this work, we first tune t_R for a high-fidelity shuttling and then tried making t_L as large as possible while keeping t_R . This limits J which results in a slow CZ gate (Supplementary Note 1). An additional barrier gate pulse would improve the gate time and thereby the fidelity in the present device, but we cannot estimate how much the fidelity increases because the dephasing during the controlled-phase accumulation depends on the gate voltage condition. To explain the above, we have modified the

manuscript as follows:

Lines 158-166:

The obtained fidelity is mostly limited by dephasing due to the slow controlled-phase accumulation of $0.4 \mu\text{s}$ compared to the decoupled dephasing times of $\sim 7 \mu\text{s}$ (Supplementary Fig. 8). Application of a barrier gate pulse in addition to the shuttling pulse would further improve the CZ gate fidelity by increasing J around the charge-symmetry point (Supplementary Note 1). In addition, the fluctuations of EDSR resonance frequencies during the data acquisition contribute to the obtained infidelity of the CZ gate. We calibrate these parameters in every ~ 2 hours and the total data acquisition takes ~ 10 hours. More frequent auto-calibration during the measurement³⁷ would further improve the gate fidelity.

Lines 337-339:

The decoupled dephasing time of Q_M is obtained from the exponential decay of the oscillation amplitude of spin-up probability as a function of the phase of the final $\pi/2$ rotation for Q_M (Supplementary Fig. 8).

In addition, we have updated decoupled dephasing times in Fig. 3d which are now obtained from the exponential decay (there were no significant changes from the original data which were obtained from a fit the data to $a \times \exp(-t_{\text{evol}}/T_{2,\text{DCZ}}^*)^n$) that includes n as a fitting parameter) and added Supplementary Fig. 8 showing a decay property of the decoupled controlled-phase accumulation.

Comment #3

I note that even if coherent errors account for a small fraction of the total error rate, there can be a significant difference between the worst-case infidelities and the average-case infidelity [see, e.g., J. Wallman et al., PRA 94, 052325 (2016) and A. Hashim et al., PRX 11, 041039 (2021)]. The authors write, “We note that the phase of Q_M shifts when it is shuttled across dots with different Zeeman energies, but this phase shift can be removed by a phase gate implemented by software”. I guess this is an attempt to avoid/suppress coherent errors; is that correct? I think in the present study, the difference between Zeeman energies across the dots is (largely) due to the B-field gradient. However, I think it is worth mentioning that change of the Si/SiGe interface roughness across different dots also contributes to this due to the change of the valley splitting as well as change in the effective strength of the spin-orbit interaction [see, e.g., A. Hosseinkhani et al., PRB 104, 085309 (2021)].

Response: We thank the reviewer for the useful comment. For the first point, yes, we compensate

coherent phase accumulations on Q_M (coherent errors) by phase gates implemented as phase shifts in subsequent microwave control pulses [e.g., refs 1,12,37]. As long as the phase accumulations are deterministic, they can be compensated for up to the precision of calibration. For the second point, we agree with the reviewer that a change of the Si/SiGe interface roughness across different dots can contribute to the Zeeman energy difference. To comment on these points, we have modified the manuscript as follows:

Lines 120-125:

We note that the phase of Q_M **coherently** shifts when it is shuttled across dots with different Zeeman energies **that originate from the micromagnet-induced gradient field and a change in the interface roughness of the heterostructure across dots**³⁶. Since t_R is sufficiently large for adiabatic shuttling of Q_M , **this phase shift is a deterministic coherent phase shift which** can be removed by a phase gate implemented by **shifting phases of subsequent control microwave pulses in zero gate time**^{1,12,37}.

References:

36. Hosseinkhani, A. & Burkard, G. Relaxation of single-electron spin qubits in silicon in the presence of interface steps. *Phys. Rev. B* **104**, 085309 (2021).

Response to Reviewer #2:

Comment #1

The paper by Noiri et al. describes the implementation of a shuttling-based exchange gate protocol, based on a triple quantum dot sample in silicon and the use of a technique to swap spin states between neighboring quantum dots. Quite a few of the techniques used in this paper have recently been introduced by the same authors in Ref. 1.

Response: Although we use the same device used in ref. 1, there are advances in this work from ref. 1. The main results of our work, simultaneous high-fidelity single-qubit gates for the local qubits with negligible exchange interaction between them, the shuttling-based efficient exchange switching, and the shuttling-based CZ gate are all new techniques we demonstrate for the first time. This is indeed pointed out in Comment #1 of Reviewer #1.

Comment #2

The manuscript is detailed and well written, the analysis of the measurements is sound, and the described experiments constitute a step into a promising direction. Especially, the “more than one

thousand switching ratio of J ” obtained is very impressive. I also agree with the authors that this method of switching the exchange gate on and off may be generally useful when they say “shuttling-mode exchange switching is beneficial for local qubit operations” and they should probably highlight this point earlier in the paper.

Response: We thank the reviewer for carefully reading the manuscript and figuring out the importance of our shuttling-based approach for scaling up. We also thank the reviewer for correctly understanding the advantage of shuttling-based efficient exchange switching. To clarify the advantage, we have modified the manuscript as follows:

Lines 60-67:

Initially, the local and moving qubits Q_L and Q_M are in the left and right dots, respectively, where parallel quantum processing with simultaneous single-qubit gates is performed. We refer to this configuration as the sparse state (Fig. 1b). **The negligible coupling between the qubits enables us to maintain the high fidelity of single-qubit gates while driving both qubits simultaneously.** To perform a two-qubit gate, Q_M in the right dot is shuttled to the center dot, and at the same time, the exchange coupling is turned on. We refer to this state as the coupled state (Fig. 1b). **The shuttling-mode exchange switching allows us to efficiently control exchange coupling with an on/off ratio above 1,000.**

Comment #3

I am however not completely convinced that the experiment at its current stage constitutes an actual demonstration of “the CZ gate between silicon spin qubits at distant quantum processors” as the authors claim in the conclusion.

A weakness of the manuscript is that the authors select an absolute minimal configuration to demonstrate the technique, namely a linear triple quantum dot where they shuttle back and forth. Using a series of empty dots as has been demonstrated previously (Mills. et. al., Ref. 26) would have added to the impact and solidified claims about the preservation of the gate fidelity over larger distances and a more varied and realistic disorder landscape. The first panel of Figure 1, a schematic, especially seems to make explicitly the claim of a long length channel, while the experiment itself is in a linear triple dot, on which similar experiments of similar or greater complexity have been performed previously. For example, the navigation in the gate spaces of longer quantum dot chains could in itself constitute a significant challenge to the protocol and I would therefore suggest shuttling over an empty chain of three to four dots would perhaps have been an actual minimal example or demonstration.

Response: As pointed out, a practical quantum link between distant quantum processors consists of a shuttling of Q_M across a long shuttling channel and the shuttling-based CZ gate as shown in Fig. 1a. Since the former technique and physics behind the coherent shuttling were explored [e.g., refs. 25-27] and are being explored by other groups, we focused on the first demonstration of the latter technique in the present study. For this purpose, our experimental setup is a minimum but sufficient one. At the same time, we agree with the reviewer that there are remaining hurdles to be overcome to implement a shuttling-based quantum link in a practical manner [e.g., over a distance of $\sim 10 \mu\text{m}$, ref. 15]. As the reviewer suggested in Comment #10, we have massively revised the manuscript to clarify the concept of linking qubits at distant quantum processors, what we have focused on and demonstrated experimentally in this work, and the remaining hurdles to realize the concept in a practical manner as follows:

Lines 23-31:

Here we **demonstrate a two-qubit gate between** spin qubits via coherent spin shuttling, **a key technology for linking distant silicon quantum processors**. Coherent shuttling of a spin qubit enables efficient switching of the exchange coupling with an on/off ratio exceeding 1,000, while preserving the spin coherence by 99.6% for the single shuttling between neighboring dots. With this shuttling-mode exchange control, we demonstrate a two-qubit controlled-phase gate with a fidelity of 93%, assessed via randomized benchmarking. **Combination of our technique and a phase coherent shuttling of a qubit across a large quantum dot array will provide** feasible path toward a quantum link between distant silicon quantum processors, a key requirement for large-scale quantum computation.

Lines 47-50:

Here we **propose and demonstrate a shuttling-based** two-qubit gate **which plays a key role in a quantum link** between distant **silicon** quantum processors by electron shuttling. Figure 1a illustrates how this technique **along with a coherent shuttling across a quantum dot array^{26,27}** can be used to interconnect two distant quantum processors via an empty quantum dot array, making a quantum link between them.

Lines 56-57:

This technique **will** enable to implement the two-qubit gate between qubits at distant quantum processors **when combined with shuttling across** a long channel.

Lines 59-60:

The experiment is performed in a tunnel coupled triple quantum dot hosting two qubits, a minimum setup to demonstrate the **shuttling-based two-qubit gate**.

Lines 68-80:

Practically, a quantum link that can couple qubits separated by $\sim 10 \mu\text{m}$ distance is useful for scaling up¹⁵. Along with the shuttling-based CZ gate, this requires high-fidelity coherent shuttling across a large quantum dot array. With a sufficiently large inter-dot tunnel coupling, we demonstrate that 99.6% of the spin phase coherence is preserved in a single shuttling cycle. Then, challenges to be overcome include precise control of a large quantum dot array. A virtual gate technique is useful for tuning up such a quantum dot array in a scalable manner^{25,31}. Furthermore, a recent demonstration of conveyor-mode shuttling³² can decrease the number of control signals in a long-distance shuttling. In this approach, a qubit is shuttled by an electrostatically defined travelling potential created by an array of gate electrodes which are connected to one of the four control signal sources. Then, the number of control signals is independent of the length of shuttling channel, potentially reducing the complexity of controlling a long shuttling channel. With such technical advances, our technique can implement a quantum link between spin qubits at distant quantum processors that is useful for scaling up.

Lines 185-192:

In summary, we demonstrate a CZ gate between silicon spin qubits by coherent shuttling of one of the qubits for linking distant quantum processors. The coherent shuttling allows us to shuttle a qubit while preserving its spin phase by 99.6% and simultaneously switch on and off the exchange coupling. The shuttling-mode exchange switching allows us to implement the CZ gate with a fidelity of 93% accompanied with a high on/off ratio of more than one thousand. Even higher gate fidelity will be achieved by an additional barrier gate pulse. These results demonstrate key technologies for a shuttling-based quantum link between distant quantum processors and thereby open a path to realization of large-scale spin-based quantum computation.

References:

31. Volk, C. *et al.* Loading a quantum-dot based ‘Qubyte’ register. *npj quantum Inf.* **5**, 29 (2019).
32. Seidler, I. *et al.* Conveyor-mode single-electron shuttling in Si/SiGe for a scalable quantum computing architecture. *arXiv:2108.00879* (2021).

Comment #4

The dependence of the spin preservation fidelity on length and magnetic field gradient inhomogeneities from the cobalt micromagnet would also have been useful to see, and assess. The authors acknowledge that the “the phase of QM shifts when it is shuttled across dots with different Zeeman energies” but they dismiss it as “this phase shift can be removed by a phase gate implemented by software”. It is not clear how much of a roadblock, or an overhead, this is to the protocol and it would be nice to get a comment from the authors on this.

Response: We thank the reviewer for the comment. We tune the inter-dot tunnel coupling t_R large enough to shuttle Q_M adiabatically between the dots. In this case, the phase shift during the shuttling is a deterministic coherent one which can be compensated by a single-qubit phase gate up to the precision of calibration. We can implement single-qubit phase gates acting on Q_M (and similarly on Q_L if needed) with arbitrary angles by shifting the phase of subsequent control microwave pulses [e.g., refs. 1, 12, 37]. They are implemented in zero-gate time, therefore adding no overhead to the protocol. To explain this, we have changed the manuscript as follows:

Lines 120-125:

We note that the phase of Q_M **coherently** shifts when it is shuttled across dots with different Zeeman energies **that originate from the micromagnet-induced gradient field and a change in the interface roughness of the heterostructure across dots**³⁶. Since t_R is sufficiently large for adiabatic shuttling of Q_M , **this phase shift is a deterministic coherent phase shift which** can be removed by a phase gate implemented by **shifting phases of subsequent control microwave pulses in zero gate time**^{1,12,37}.

Comment #5

1) Could the authors comment on the effect of gate crosstalk on the fidelity of this protocol, especially if other qubits are around, or if the shuttling needs to be performed on longer length scales? I would have expected this to be something that needs to be compensated for, especially with the overlapping-gate geometry.

Response: We assume that “gate crosstalk” in this comment is capacitive crosstalk in gate electrodes (not the one in quantum gates). Ideally, shuttling between two quantum dots in a large array should occur without affecting the other, stationary quantum dots. However, the crosstalk between gates may cause unwanted charge transitions or phase shifts in the stationary quantum dots. Although we did not use in this work, a well-established virtual gate technique [e.g., refs. 25, 31] could be applied to avoid these unwanted effects by the gate crosstalk. This technique will be required when moving a qubit across a large array of quantum dots in future experiments. To comment on the above, we have modified the manuscript as follows:

Lines 72-74:

Then, challenges to be overcome include precise control of a large quantum dot array. A virtual gate technique is useful for tuning up such a quantum dot array in a scalable manner^{25,31}.

References:

31. Volk, C. *et al.* Loading a quantum-dot based ‘Qubyte’ register. *npj quantum Inf.* **5**, 29 (2019).

Comment #6

2) The authors also comment, if I understand right, that the high fidelity of single qubit gates can be maintained without making corrections for cross talk when the qubits are in the “sparse” position? They say “with this condition, we also obtain a high-fidelity ($F = 99.906 \pm 0.002\%$ for Q_L and $99.751 \pm 0.003\%$ for Q_M) single-qubit gates (Fig. 1f, g) with a Rabi frequency of 2.5 MHz even when the same gate sequence is applied to both qubits simultaneously”.

Response: Yes, the single-qubit gate fidelities for Q_L and Q_M are 99.91% and 99.75%, respectively when they are simultaneously operated in the sparse state without making any corrections for single-qubit gate cross talk. Specifically, we argue that the following crosstalk errors are negligible in the sparse state: the off-resonant driving due to overlap of the EDSR conditions, the resonance frequency shift under microwave drive [e.g., refs. 2, 39], and the effect of residual exchange coupling. The off-resonant driving is completely negligible as the difference in the resonance frequencies of Q_L and Q_M (733.4 MHz) in the sparse state is much larger than their Rabi frequencies of 2.5 MHz (resonance widths are determined by the Rabi frequency). This is quite easily achieved using the micromagnet. The microwave-induced frequency shift could be significant with a larger microwave power, but we chose small enough power such that it is negligible. In our experience, this effect is relatively weak in a device with overlapping gate structure perhaps due to a strong confinement. The effect of the exchange coupling is also negligible (see Supplementary Note 2 and Supplementary Fig. 9) thanks to the residual coupling as small as 0.9 kHz at the sparse state (Supplementary Fig. 6b). To clarify the motivation of the simulation in Supplementary Fig. 9, we have modified the manuscript as follows:

Lines 29-34 in Supplementary Information:

Supplementary Note 2: Simulation of EDSR frequency detuning of single-qubit gate fidelity

We simulate the effect of residual exchange coupling on the single-qubit primitive gate fidelity as it is one of the most relevant sources of a gate crosstalk¹⁻³ which needs to be avoided for scaling up. Under a finite exchange coupling, resonance frequency of a qubit depends on the state of the other qubit, making a drive of single-qubit gate slightly off-resonant. Therefore, we discuss how the single-qubit gate fidelity depends on EDSR frequency detuning in this section.

Comment #7

3) The manuscript also mentions that “Although the device is identical to the one used in ref. 1, we obtain a shorter dephasing time $T2^*$ for both qubits than those measured in ref. 1 possibly due to increased charge noise in the gate voltage condition used in this work”. It is not clear if the

decreased T_2^ relates to the measurement in the sparse state or is simply due to changes in the experimental conditions between the two experiments; can the authors comment on this?*

Response: Shorter T_2^* observed in this work compared to ref. 1 does not relate to the sparse state but due to changes in the experimental condition. Spin-electric coupling of a spin qubit is influenced by the orbital splitting energy and a field gradient by the micromagnet [e.g., ref. 5]. Activation of charge impurities also depends on gate voltage conditions. Both effects give rise to gate-voltage-dependent T_2^* in ^{28}Si spin qubit devices. In the particular device used here, we phenomenologically observe that T_2^* decreases when we increase the barrier gate voltages to enhance t_L and t_R . While the experimental limitation (pulse lines not connected to the barrier gates) necessitates the use of a voltage condition with large t_L and t_R , for example, we measure much longer T_2^* in the sparse state with decreased barrier gate voltage for both qubits when t_L and t_R are small (Supplementary Fig. 3). To explain this point, we have added Supplementary Fig. 3 which has been cited in the caption of Supplementary Fig. 2.

Comment #8

4) The paragraph (line 126) describing CZ gate implementation via a DCZ gate (and especially the equivalence via a Y and Z/2 gate) is not clear, and could be explained in a little more detail to convince the reader that the CZ is appropriately performed.

Response: We thank the reviewer for the comment. We agree that more detailed explanation is helpful to convince the readers. During the preparation of the revised manuscript, we became aware that the DCZ gate used for producing Fig. 4f is slightly different from that used in Fig. 4a and therefore the original explanation for constructing the CZ gate from the DCZ gate was wrong. We have corrected this point and added Fig. 4c which has been cited at the line 149.

Comment #9

5) Was there a problem (device or experimental) due to which a symmetric or barrier-gate pulse was not used to operate the CZ gate (line 146) as the authors mention?

Response: As we wrote in the original manuscript, barrier gate pulses would improve the CZ gate fidelity, but we could not use them in this work due to the following two reasons. The main limitation is the arbitrary wave form generator used in the experiment, which has only four outputs connected to the three plunger gates P1, P2, and P3, and the sensor plunger gate (see Measurement setup in Methods). In addition, the exchange controllability by the barrier gate is low in this device due to the design of gate electrodes (see Supplementary Note 1). To explain the above, we have modified the manuscript as follows:

Lines 18-25 in Supplementary Information:

The small (~ 25 nm) gap between the plunger gates makes the control of inter-dot tunnel coupling by the barrier gate inefficient and a large (> 1 V) positive voltage is required to achieve sufficiently large inter-dot tunnel couplings t_R and t_L **simultaneously** for high-fidelity electron shuttling and for inducing a large J . We also find that the device becomes unstable if the barrier gate voltage exceeds 1 V, which limits the available range of J in this device. **We note that barrier gate pulses reduce the requirement of making t_R and t_L large simultaneously, but we cannot use them in this work due to the limitation of the number of outputs of the arbitrary waveform generator used (see Methods).**

Comment #10

To conclude, the work is at a very high level and is very useful to the community; however due to the inequivalence of shuttling within a triple dot vs shuttling along longer distances, it is not clear to me that the claim of a long-range gate should be made. I feel that claim should be reserved for an actual demonstration of a long-range gate. Perhaps the paper can still be suitable for publication in this journal, given a change in the sections of the manuscript where authors have made claims of this type, such as “Here we overcome this problem by coupling spin qubits at distant quantum processors”, for example on line 23 and in the conclusion.

Response: We thank the reviewer again for carefully reading the manuscript and finding it high quality and useful to the community. In response to the reviewer’s comments, we have massively modified the manuscript to clarify the concept of linking qubits at distant quantum processors, what we have focused on and demonstrated experimentally in this work, and the remaining hurdles to realize the concept in a practical manner (see also response to Comment #3). We believe that the main concern of the reviewer has been addressed in the revised manuscript.

Comment #11

Lastly, a few typos:

Concluding paragraph: “we demonstrate a CZ gate” followed by “the coherent shuttling allows us to shuttle a qubit while preserving its spin phase by 99.6%”

Fig. 1 caption: “Concept of experiment and qubit characterization”

Response: We thank the reviewer for pointing out typos. We have modified the manuscript as follows:

Lines 185-187:

In summary, we demonstrate a CZ gate between silicon spin qubits by coherent shuttling of one of the

qubits for linking distant quantum processors. The coherent shuttling allows us to shuttle a qubit while preserving its spin phase by 99.6% and simultaneously switch on and off the exchange coupling.

Fig. 1 caption:

Fig. 1 Concept of experiment and qubit characterization.

Response to Reviewer #3:

Comment #1

Summary of the work

In this work, A. Noiri and colleagues demonstrate a coherent shuttling-based controlled phase (CZ) gate between two one-site-apart single-spin qubits (Loss-DiVincenzo qubit) in a triple quantum dot device in 28Si/SiGe. Specifically, they show coherent shuttling of an electron between the center (dot2) and right sites (dot3) by showing both transfers of basis and superposition states. They also implement coherent shuttling + echoed CZ gate with a fidelity of 93% deduced from two-qubit Randomized benchmarking protocol.

Evaluation

1. Strengths: I acknowledge the work's high-quality results. The authors' experiment is from one of the best quality silicon quantum information devices available to date. All single and two-qubit gate fidelity, coherent shuttling performance, and full two-qubit randomized benchmarking represent the state-of-the-art of the relevant research field. The manuscript is concise and generally well written except for a few points I describe below.

Response: We thank the reviewer for carefully reading the manuscript and finding it high quality.

Comment #2

2. Clarification on novelty: Although the authors correctly cite previous demonstrations, I want to clarify that the demonstration of shuttling + CZ gate in combination is, and only this is, the novel aspect of the paper. Individual demonstration of coherent shuttling (both basis and superposition states) in 28Si, high fidelity single-spin qubit gates, and echoed CZ gate and two-qubit Randomized benchmarking are all demonstrated either by other groups or by the authors previously. As such, the work's evaluation should be based on whether the current demonstration of shuttling + CZ gate indeed has a significant and promising impact on the long-range linking of qubits in distant

quantum processors.

Response: We would like to stress that our demonstration is not just a combination of already demonstrated coherent shuttling and CZ gate. We develop a new exchange switching technique (additionally, we used detuning control to further enhance exchange coupling) by coherent shuttling to implement our CZ gate meaning that the coherent shuttling and the exchange switching are implemented at the same time. This is different from all of the reported CZ gates where the exchange couplings are controlled by energy detuning and/or potential barrier between quantum dots. This point is indeed mentioned in Comment #1 of Reviewer #1 and Comment #2 of Reviewer #2. Since our shuttling-based CZ gate plays a key role for making shuttling-based quantum link, we believe that our work provides a significant advance in the field toward long-range linking of qubits in distant quantum processors. To clarify the above point, we have modified the manuscript as follows:

Lines 54-56:

In contrast to previous demonstrations of a CZ gate^{2,3,6,13}, our two-qubit gate between the local and moving qubits relies on dynamical switching of the exchange coupling by the shuttling processes.

Comment #3

3. Weakness: I have main concerns as follows.

A. Major point: A credible (through RB) evaluation of 93% fidelity of shuttling-based CZ gate itself is not bad at all considering that it is the first demonstration of this kind. However, the work's main aim is to use coherent shuttling for linking distant qubits. Thus, in my opinion, a more correct evaluation should be as follows.

i. First perform conventional CZ gate between dot1 and dot2. That is conventional CZ when the electrons are sitting at dot1 and dot2 without shuttling. And evaluate through two-qubit RB.

ii. Then perform shuttling + CZ gate between dot1 and dot3 and do two-qubit RB. This is already demonstrated in the current work.

iii. Repeat (ii) with more than one shuttling before CZ. Compare the RB results with the result in (i), and concretely evaluate how much the shuttling process affects the reduction of CZ fidelity.

In short, my main concern is the absence of experiments A-(i) and A-(iii). For example, if I use the conventional CZ (coupled state CZ) fidelity of 99.5% previously demonstrated by the authors

(Nature 601 338) in the same device but between dot2 and dot3, I should conclude that one shuttling process reduces CZ gate fidelity by $93/99.5 = 93.4\%$, which means the CZ fidelity drop down to 70% with only five shuttling process. This is contrary to the authors' main claim; coherent shuttling is not promising for connecting distant qubits!

Response; As explained in our response to Comment #2, we cannot simply separate our CZ gate into shuttling + controlled-phase accumulation. Therefore, the proposed protocol (i)-(iii) cannot be applied to assess our CZ gate. We agree with the reviewer that adding more information about the error sources of our CZ gate is valuable for readers to understand the CZ gate performance. We argue that the infidelity of the present experiment seems to be mainly due to the dephasing effect during the CZ gate time rather than those during the shuttling process for the following reasons. First, the dephasing noise leads to an exponential decay in the controlled-phase accumulation with the sequence shown in Fig. 3c (see Supplementary Fig. 8) [e.g., ref. 13], from which we obtained the decoupled dephasing times of ~ 7 and $10 \mu\text{s}$ for Q_L and Q_M , respectively. The infidelity during the controlled-phase accumulation therefore amounts to $1 - e^{-(0.4/7)} = 5.5\%$. This suggests the most part of the error in our CZ gate comes from dephasing during the controlled-phase accumulation. Secondly, the shuttling fidelity (which accompanies switching of the exchange coupling) is separately evaluated and found to be high as demonstrated in Fig. 2. Although we only measure the coherence preservation of Q_M during shuttling cycles, reasonable error mechanisms in the two-qubit Hilbert space, if any, would affect the result of this measurement. The observed coherence of Q_M suggests that errors in the shuttling process are small.

Overall, the fidelity of shuttling-based CZ gate in the particular device condition used in this work seems to be mostly limited by the dephasing during controlled-phase accumulation (see also response to Comment #2 of Reviewer #1) and the shuttling fidelity is high (99.6% per shuttling cycle). We note that the CZ gate fidelity obtained in this work and the fidelity reported in ref. 1 cannot be directly compared. This is because in ref. 1, we implemented a resonant CNOT gate under fixed exchange coupling (not a CZ gate) in a significantly different device condition (operation at the charge symmetry point with a large $J = 19 \text{ MHz}$). If we compare the dephasing effect, it is much stronger in this work than that of ref. 1. For example, the gate time in this work is four times longer than that of ref. 1 for similar dephasing times (decoupled dephasing times in this work). In addition, dephasing noise is somewhat decoupled by the resonant CNOT gate used in ref. 1, while it directly leads to dephasing during the controlled-phase accumulation in this work. To add a comment on the dephasing during controlled-phase accumulation, we have added Supplementary Fig. 8 which has been cited at the line 160.

Comment #4

B. Major point: Related to A, the authors stress that the shuttling is beneficial for increasing on/off ratio of J . I agree with the authors. However, without the experiment described in A-(i) and (iii), the current paper's utility is rather for showing enhanced controllability over J through shuttling than using shuttling for connecting distant qubits. Note also that some of the barrier gates in the current device have limited J controllability. If the authors could manage to build a device with increased interdot distance, I would expect the same J controllability can be achieved without shuttling, making advantage of increasing J controllability with shuttling technique marginal.

Response: As we explained in response to Comments #2 and #3, the main scope of this work is to develop shuttling-based CZ gate which plays a key role in shuttling-based quantum link and improvement of controllability of J is just an additional advantage. We agree with the reviewer that the recent technical advances have improved exchange coupling controllability by barrier gate pulse [e.g., ref. 46]. Our exchange switching technique can be used together with a barrier gate exchange control to further enhance the exchange controllability so that our technique might be still beneficial for a device having a good barrier gate controllability. This is because a larger on/off ratio of the exchange switching is better to achieve high-fidelity single- and two-qubit gates at the same time (see Supplementary Note 2). For instance, single-qubit gate fidelity above 99.999% is possible with on/off ratio of 10,000 (with $J \sim 10$ MHz, the residual coupling of ~ 1 kHz, and single-qubit Rabi frequency of 2.5 MHz) if we only consider the effect of the residual coupling. Such a high on/off ratio is still challenging to achieve by a barrier gate pulse even with a current state-of-the-art device [e.g., ref. 46]. Therefore, our exchange switching technique might be useful to further improve gate fidelities in future experiments. To explain this point, we have modified the manuscript as follows:

Lines 178-183:

We note that controllability of the coupling by the conventional schemes has been improved recently to the on/off ratio of 1,000 in an advanced device structure⁴⁶ but an even larger on/off ratio may be required for further enhancing the gate fidelity. The shuttling-mode exchange switching can be used together with the conventional technique to improve the exchange controllability and thus favorable not only for linking distant quantum processors but also for implementing high-fidelity local qubit operations.

Comment #5

C. Minor point: Coherent shuttling for linking distant qubits is a neat concept, but it does not reduce the complexity of the device architecture: for linking, one still needs to fabricate and use many control lines synchronously. Can authors put their opinion on this and discuss briefly?

Response: We thank the reviewer for the comment. Recently a clever idea and its initial proof-of-principle experiment have been demonstrated for reducing complexity of device requirements for a large shuttling channel [ref. 32]. In this approach, a qubit is shuttled by a travelling potential rather than shuttling between nearest-neighbor quantum dots one by one. The travelling potential is created by sinusoidal voltages applied to an array of gate electrodes which are connected to one of the four control signal sources. Importantly, the number of control signals is independent of the distance of shuttling channel although the number of gates increases with distance. With the technique, a long-distance quantum link between silicon quantum processors will be realized without requiring large number of control lines. To comment on this, we have modified the manuscript as follows:

Lines 74-79:

Furthermore, a recent demonstration of conveyer-mode shuttling³² can decrease the number of control signals in a long-distance shuttling. In this approach, a qubit is shuttled by an electrostatically defined travelling potential created by an array of gate electrodes which are connected to one of the four control signal sources. Then, the number of control signals is independent of the length of shuttling channel, potentially reducing the complexity of controlling a long shuttling channel.

References:

32. Seidler, I. *et al.* Conveyer-mode single-electron shuttling in Si/SiGe for a scalable quantum computing architecture. *arXiv:2108.00879* (2021).

Comment #6

D. Major-Minor mixed: I see in the Extended data 2 that T_2^ in the current device (28Si) is only marginally improved over the natural Si and authors ascribe to increased charge noise. Can authors put a bit more discussion on this? Why charge noise affects nominally charge-noise immune single spin qubits? Is it related to the interplay between increased charge noise and micromagnet? Related to my Major points, is this the reason that shuttle+CZ gate in the current work has lower fidelity 93% than Ref. 1 ? Not because of the shuttle process itself?*

Response: As pointed out, single spin dephasing is caused by charge noise via a field gradient by the micromagnet. Charge noise of quantum dot devices and sensitivity of qubits to the noise are known to depend on gate voltage conditions. Both effects give rise to gate-voltage-dependent T_2^* in ²⁸Si spin qubit devices. In the particular device used here, we phenomenologically observe that T_2^* decreases when we increase the barrier gate voltages to enhance t_L and t_R . While the experimental limitation (pulse lines not connected to the barrier gates) necessitates an appropriate gate voltage condition to keep large t_L and t_R , for example, we measure much longer T_2^* with decreased barrier gate voltage for

both qubits when t_L and t_R are small (Supplementary Fig. 3). As in the response to the previous comment, we believe that the main limitation in the CZ gate in this work is the dephasing during the relatively long gate time and short DCZ decay time. To explain this point, we have added Supplementary Fig. 3 which has been cited in the caption of Supplementary Fig. 2.

Comment #7

Overall, I acknowledge the work's highly qualified experimental results and the importance of linking distant qubits in the future. However, the current version of the manuscript lacks an important series of experiments as I described in 3.A. I understand that performing one two-qubit RB protocol takes a lot of time and there can be device-specific problems (barrier controllability, charge stability.. etc.) that prevents the execution of some of the additional experiments I suggest. However, these experiments are crucial for this paper, especially for talking about quantum links even though it is performed in a minimal set of triple quantum dots. Therefore, I would like to postpone my recommendation until I get a reply from the authors with some additional evidence or rebuttal to my points.

Response: We thank the reviewer again for corroborating the high quality of our experimental results and importance of linking distant qubits. As explained above, we believe that the revised manuscript has addressed all the concerns of the reviewer.

REVIEWERS' COMMENTS

Reviewer #1 (Remarks to the Author):

The authors have taken my comments into consideration and revised the manuscript accordingly. I believe the paper can be published now.

Reviewer #2 (Remarks to the Author):

I have now read the resubmitted manuscript and the authors' response, and I have a greater understanding of the points I had raised in my previous report. I appreciate the detailed answers to my questions, taking on board of the comments, extensive clarifications in the manuscripts and the toning down of some of the language via the changes made to the introduction and conclusion. I am also glad that Comment 8 about the DCZ was helpful to the authors. I also appreciate that in response to Reviewer 3's comments, they have acknowledged recent work in conveyor mode shuttling. The manuscript I think is currently valuable to the community and in its current form, and useful as a publication in Nature Communications. The one reservation I still have is to what extent a shuttling gate carried out within the same two quantum dots can truly be referred to as a long-distance link. Indeed, from the current manuscript it is clear that it can be such a long-distance gate, but it is unclear to me if this current work is a full experimental demonstration of this long-range gate. However, this is an editorial decision as to impact, and as such as far as my remit as a reviewer goes, I would be okay with recommending publication in Nature Communications.

Reviewer #3 (Remarks to the Author):

The authors have made a serious effort to improve the manuscript and adequately addressed the comments from the reviewers. On one hand, I still think that it is desirable to perform experiments using a shuttling-based CZ gate with more than a single shuttling, especially in order to discuss the true potential of the technique. After all, many shuttling processes would be needed in the future in a longer quantum dot chain.

On the other hand, now I understand and agree with the authors that the current demonstration is not just CZ + shuttling, and rather than the experiment I proposed previously, the authors' assessment of limiting factors of fidelity of CZ gate is more appropriate, at least for currently minimum setting of triple quantum dots. The added explanation of dephasing during the phase accumulation helps the reader to understand what the next improvement should be made. Overall, the current demonstration itself is the first of its kind. Thus, although it is performed in an absolutely minimum setting, the work has a significant enough impact on the broad readership of the journal.

Also, the authors properly emphasized the importance of enlarging the on/off ratio of J, through the combination of the shuttling technique and barrier gate control to realize two-qubit fidelity significantly higher than before. Replies to my minor comments are also reasonable and I agree with them.

With this, I now recommend the manuscript for publication in Nature Communications as-is. I have no further comments.